# Synaptic Dynamics Realize First-order Adaptive Learning and Weight Symmetry

## Abstract

Gradient-based first-order adaptive optimization methods such as the Adam optimizer are prevalent in training artificial networks, achieving the state-of-the-art results. This work attempts to answer the question whether it is viable for biological neural systems to adopt such optimization methods. To this end, we demonstrate a realization of the Adam optimizer using biologically-plausible mechanisms in synapses. The proposed learning rule has clear biological correspondence, runs continuously in time, and achieves performance to comparable Adam's. In addition, we present a new approach, inspired by the predisposition property of synapses observed in neuroscience, to circumvent the biological implausibility of the weight transport problem in backpropagation (BP). With only local information and no separate training phases, this method establishes and maintains weight symmetry in the forward and backward signaling paths, and is applicable to the proposed biologically plausible Adam learning rule. The aforementioned mechanisms may shed light on the way in which biological synaptic dynamics facilitate learning.

## 1 Introduction

Gradient-based adaptive optimization is a widely used in many science and engineering applications including training of artificial neural networks (ANNs) (Rumelhart et al., 1986). In particular, first-order methods are preferred over higher-order methods since their memory overhead is significantly lower, considering ANNs are often characterized by high dimension feature spaces and large numbers of parameters. Among the most well-known ANN training methods are stochastic gradient descent (SGD) with momentum (Rumelhart et al., 1986), root mean square propagation (RMSProp) (Tieleman & Hinton, 2012), and adaptive moment estimation (Adam) (Kingma & Ba, 2014). Different from gradient descent, which optimizes an loss function over the complete dataset, SGD runs on a mini-batch. The momentum term accelerates the adjustment of SGD along the direction to a minima, and RMSProp impedes the search in the direction of oscillation. The Adam optimizer can be considered as the combination of the above two ideas; and it is computationally efficient, has fast convergence, works well with noisy/sparse gradient, and achieves the state-of-the-art results on many AI applications (Dosovitskiy et al., 2020; Wang et al., 2022).

Given the success of gradient-based adaptive optimization techniques, particularly Adam, in training ANNs, it is natural to ask the question whether it is viable for biological neural systems to adopt such optimization strategies. We attempt to answer this question by demonstrating an implementation of the Adam optimizer based on biologically plausible synaptic dynamics and a new solution to the well-known weight transport problem. We call our implementation Bio-Adam.

Nevertheless, it is not immediately clear how to realize the Adam optimizer biologically realistically given its intricacies. Comparing to the classical SGD method, Adam has two major new ingredients: use of momentum $m$ to smooth the gradient $g$ of multiple batches, and division by a smooth estimation $\sqrt{v}$ of the root mean square of the gradient to constrain the step size, which is also known as the RMSProp term $1/(\sqrt{v}_t + \epsilon)$, in which $\epsilon$ is a small number to prevent division by zero. Although signal smoothing is commonly done in biological modeling such as by using the leaky-integrate and fire (LIF) model of spiking neurons (Gerstner et al., 2014), we identify that the root mean square calculation $\sqrt{v}$ and the existence of the division operator in Adam are biologically problematic. With respect to these difficulties, we define a new variable $\rho$ to mimic the dynamics of RMSProp.

Biologically, this newly defined $\rho$ variable may be thought as the concentration of certain synaptic substance consumed during the learning process. Essentially, $\rho$ is designed based on the following biologically motivated ideas. A large weight updating signal accelerates the consumption of the substance, dropping the concentration of the substance, which in turn slows down the pace of weight update. On the other hand, under a small weight updating signal, the synapse gradually replenishes the substance and restores the fast weight update pace. This kind of behavior mimics the RMSProp term $1/(\sqrt{v}_t + \epsilon)$ in Adam. The overall weight update speed of our biological Adam is proportional to the product of $m$ and $\rho$.

Another roadblock to a biologically plausible realization of Adam is the weight transport problem, which is in fact an issue common to the wider family of all BP methods. BP is generally believed to be biologically implausible in the brain for several reasons (Stork, 1989; Illing et al., 2019). One key issue is that BP requires symmetrical weights between the forward and backward paths to propagate correct error information. This is also known as the weight transport problem (Grossberg, 1987; Ororbia II et al., 2017), which we address by a new approach, inspired by the predisposition property of synapses observed in neuroscience. According to predisposition (Chistiakova et al., 2014), weak synapses that have been depressed are more prone to potentiation, whereas strong ones are more prone to depression. When the forward and backward synapses share local updating signals, a potentiation signal potentiates the stronger synapses less than the weaker ones. Conversely, a depression signal depresses the weaker synapses less than the stronger ones. By leveraging the biological predisposition property, the proposed mechanism eventually aligns the forward weights with the backward weights. There are other methods also mentioned to share gradients, like the weight-mirror and the modified KP algorithm presented in (Akrout et al., 2019). Yet we believe our method is more bio-plausible for it is based on the well-observed synaptic property. Our solution to the weight transport problem is immediately applicable to the proposed Adam learning rule.

To conclude, the presented Bio-Adam learning rule is based on biologically plausible synaptic dynamics with an underlying biological mechanism addressing the weight transport problem. Moreover, our biological Adam optimizer delivers a performance level that is on a par with its original implementation (Kingma & Ba, 2014). Our findings may shed light on the way in which biological neural systems facilitate powerful learning processes.

## 2 BIO-PLAUSIBLE FIRST-ORDER ADAPTIVE OPTIMIZERS

### 2.1 PRELIMINARIES

The classical stochastic gradient descent (SGD) follows:

$$\theta_t \leftarrow \theta_{t-1} - \gamma g_t, \tag{1}$$

where $\theta$ is the weights, $\gamma$ is the learning rate, and $g$ is the gradient. SGD with momentum (Rumelhart et al., 1986) further introduced a momentum term $m$ to smooth the gradient $g$ of multiple batches according to the following dynamics:

$$m_t \leftarrow \beta_1 m_{t-1} + (1 - \beta_1)g_t, \tag{2}$$

where $\beta_1$ is a hyperparameter in the range of $[0, 1)$ and is usually close to 1.0. The final updating rule is obtained by changing $g$ to $m$ in (1), yielding: $\theta_t \leftarrow \theta_{t-1} - \gamma m_t$.

RMSProp (Tieleman & Hinton, 2012) computes a smooth estimate of the root mean square $\sqrt{v}$ of the gradient and uses it in the denominator to constrain the step size in the oscillation direction. The dynamic of $v$ follows:

$$v_t \leftarrow \beta_2 v_{t-1} + (1 - \beta_2)g_t^2, \tag{3}$$

where $\beta_2$ is a hyperparamter. Similar to $\beta_1$ in (2), $\beta_2 \in [0, 1)$ typically has a value near 1.0. The updating rule of RMSProp is: $\theta_t \leftarrow \theta_{t-1} - \gamma g_t/(\sqrt{v}_t + \epsilon)$, where $\epsilon$ is a small number to prevent division by zero.

Adam (Kingma & Ba, 2014) can be viewed as a combination of momentum and RMSProp by omitting minor adjustments such as the bias correction and use of the infinity norm. Adam's main updating rule is:

$$\theta_t \leftarrow \theta_{t-1} - \gamma m_t/(\sqrt{v}_t + \epsilon). \tag{4}$$

## 2.2 BIO-PLAUSIBLE REALIZATION

### 2.2.1 OVERALL DESIGN OF THE BIO-PLAUSIBLE SYNAPTIC DYNAMICS FOR BIO-ADAM

Molecular studies have connected synaptic plasticity to dozens of chemical and biological events, and the direction of adjustment, long-term-potentiation (LTP) or long-term depression (LTD), is a crucial determinant of whether and how chemicals are involved (Bliss & Cooke, 2011; Miranda et al., 2019; Wiera & Mozrzymas, 2021).

It is importantly to know that, when an optimizer like Adam is used, the instant adjusting signal corresponding to the negative gradient $-g$, may not be completely proportional to the actual weight update $\Delta w$. Therefore, we call the direction of the instant adjusting signal potentiation or depression, to distinguish from the LTP/LTD that actually applied on the synapse.

Our synaptic model in the proposed Bio-Adam receives two types of inputs that come from the interactions with other neurons: the potentiation signal and the depression signal, which correspond to the conditions $\{g(t)|g(t) < 0\}$ and $\{g(t)|g(t) > 0\}$, respectively. We assume that these two signals are readily available on each synapse, and refer interested readers to previous works for more discussions (Ponulak & Kasiński, 2010; Lillicrap et al., 2016; Sacramento et al., 2018; Payeur et al., 2021).

As shown in Figure 1 (A), our design utilizes two types of substances only: $m$ and $\rho$. Despite the fact that biological synapses contain various additional types of substance dynamics, the two types of substances we choose here provide enough flexibility to design dynamics akin to those of first-order adaptive optimizers.

The first type contains a group of substances that are modulated *complementarily* by the potentiation and depression stimuli. As detailed in Figure 1 (A), we name this type of substances as $m$. Further, it can be divided into $m^+$ and $m^-$ depending on whether its plasticity effect is LTP or LTD. The potentiation signal transforms $m^-$ to $m^+$, and the depression signal reversely transforms $m^+$ to $m^-$. Therefore, the total quantity of $m^+$ and $m^-$ does not change while the quantity of each changes in a complimentary way. Such complementarity is very common in biological synapses. For example, the phosphorylation and de-phosphorylation processes, controlled by potentiation and depression stimuli respectively, follow such complementary rule and apply on multiple targets such as calcium/calmodulin-dependent kinase II (CaMKII), cAMP response element-binding protein (CREB) and cofilin (Knobloch & Mansuy, 2008). Based on these biological bases, the final dynamics of $m$ in our setting exactly replicates the updating rule of the momentum term, which is the leaky-integration (exponentially smoothing) of both potentiation and depression stimuli, or the gradient per se. The dynamics of $m$ is illustrated in Figure 1 (B).

The second type of substances $\rho$ is *co-consumed* by both potentiation and depression signals. As reflected in Figure 1 (A), $\rho$ has a simple biological dynamics, which is balanced by two forces: consumption and supplementation. We assume that the consumption speed is linearly proportional to the current concentration of substances: the more abundant the substances, the faster they are consumed. In addition, we assume that the supplementation speed is linearly proportional to the difference between the saturation value $\rho_{\text{rest}}$ and the current value $\rho$: when $\rho$ is low, supplementation would be fast. While when $\rho$ reaches saturation, supplementation would slow down and finally stop. This type of dynamics refers to non-instantaneous chemical/biological processes such as translation and transcription. For example, the Brain Derived Neurotrophic Factor (BDNF) is consumed by both LTP and LTD, and supplemented through the transcription in the nucleus (Miranda et al., 2019). Interestingly, as we will show, with these chosen bio-plausible properties, $\rho$ exhibits a dynamics that is nearly identical to the RMSProp term $1/(\sqrt{v} + \epsilon)$ as depicted in the last subplot of Figure 1 (B).

In conclusion, our synaptic model has three inputs: the potentiation signal, the depression signal, and the supplementation signal. It operates on two types of substances: the complementarily modulated type $m$ and the co-consumed type $\rho$. The first two input signals correspond to the negative and positive parts of the gradient $g$, which are assumed to be readily available. The last input comes from the intraneuron dynamics. Our proposed system adopts the basic bio-plausible leaky-integrate model on both $m$ and $\rho$, and linear models are adopted to describe the consumption and supplementation speed of $\rho$. The overall weight update of the proposed Bio-Adam is proportional to the product of the density of the two types of substances $m$ and $\rho$:

$$\theta_t \leftarrow \theta_{t-1} - \gamma m_t \rho_t. \tag{5}$$

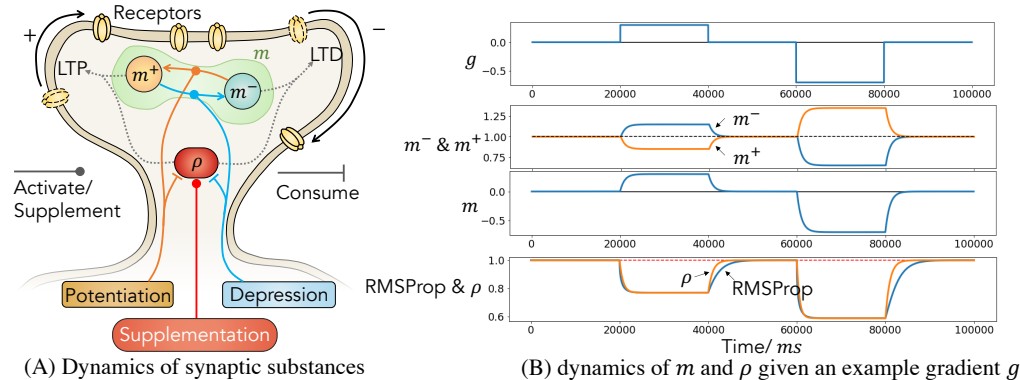

(A) Dynamics of synaptic substances    (B) dynamics of $m$ and $\rho$ given an example gradient $g$

Figure 1: First-order optimizers based on synaptic substance dynamics. (A) $m^+$ and $m^-$ are the densities of two groups of substances that modulates LTP and LTD respectively, and they are complementarily controlled by potentiation and depression stimuli. The combination of them forms the momentum term $m = (m^- - m^+)/2$. $\rho$ represents the density of the second types of substances that co-consumed by both potentiation and depression stimuli. The product of $m$ and $\rho$ affects the overall adjustment's strength and direction. (B) An example of different variables' dynamics along time is provided. Please reference the exact dynamics in Section 2.2.2 and 2.2.3. In this example, our assumed gradient is divided into five slices along the 100s total simulation time, and the values are [0, 0.3, 0, -0.7, 0] in each slide respectively. During the simulation, we set $\tau_m = \tau_\rho = 1000ms$, $\rho_{\text{rest}} = 1/\epsilon = 1$. Meanwhile, we set the resting densities of $m^+$ and $m^-$ as one. As one may observe from the bottom subplot, The proposed dynamics of $\rho$ converges to the same value as RMSProp's, and only diverges slightly in the adjusting period

Moreover, all variables in the system have their clear biological correspondences. Therefore, the system yields good bio-plausibility.

Next, we present the detailed design of the proposed biological Adam rule and analyze its dynamical behavior.

### 2.2.2 BIOLOGICALLY PLAUSIBLE REALIZATION OF MOMENTUM

The concentration of a substance is always positive. To deal with this, we implement the momentum term $m$ using two separate concentrations $m^+$ and $m^-$, which produce LTP and LTD, respectively. Additionally, we name the potentiation and depression signals as $x^+$ and $x^-$, corresponding to the absolute value of negative gradient and positive gradient:

$$x^+(t) = |g(t)| \cdot (g(t) < 0), \quad x^-(t) = |g(t)| \cdot (g(t) > 0) \tag{6}$$

Using a leaky-integration dynamics to model $m^+$ and $m^-$, we have our complementary rule as:

$$\begin{aligned}\tau_m(dm^+(t)/dt) &= -(m^+(t) - m_{\text{rest}}) + x^+(t) - x^-(t), \\ \tau_m(dm^-(t)/dt) &= -(m^-(t) - m_{\text{rest}}) + x^-(t) - x^+(t),\end{aligned} \tag{7}$$

in which $\tau_m$ is the time constant, and $m_{\text{rest}}$ is the concentration of $m^+$ and $m^-$ at rest, i.e., when no stimulus is received. Noting that $m$ in (2) is the smoothed gradient, which corresponds to the opposite of the updating direction. We define $m$ by $m = (m^- - m^+)/2$, as the overall impact of $m^+$ and $m^-$. Combining the biologically plausible dynamics of (6) and (7) gives the continuous-time dynamics of $m$, which has a form identical to the original momentum implementation of (2) (Rumelhart et al., 1986):

$$\tau_m \frac{dm(t)}{dt} = -m(t) + g(t). \tag{8}$$

The above dynamics can be discretized in time, e.g., by using the first-order forward Euler method:

$$m_t \leftarrow (1 - \frac{1}{\tau_m})m_{t-1} + \frac{1}{\tau_m}g_t. \tag{9}$$

$\beta_1$ in (2) corresponds to $(1 - 1/\tau_m)$: $\tau_m = \frac{1}{1-\beta_1}$. For example, $\beta_1 = 0.9$ correspond to $\tau_m = 10$.

**Algorithm 1** Bio-Adam, our proposed biologically plausible version of the Adam optimizer. See Section 2 for details. Good default settings for the tested machine learning problems are $\gamma = 0.0001$, $\tau_m = 10$, $\tau_\rho = 1000$ and $\rho_{\text{rest}} = 10^8$

---

**Require:** $\gamma$: Stepsize
**Require:** $\tau_m, \tau_r ho$: Time constant of $m$ and $\rho$
**Require:** $f(\theta)$: Stochastic object function with parameters $\theta$
**Require:** $\theta_0$: Initial parameter vector
   $m_0 \leftarrow 0$ (Initialize the vector of the *complementarily modulated* substances' density)
   $\rho_0 \leftarrow 1$ (Initialize the vector of the *co-consumed* substances' density)
   $t \leftarrow 0$ (Initialize timestep)
   **while** $\theta_t$ not converged **do**
      $t \leftarrow t+1$
      $g_t \leftarrow \nabla_\theta f_t(\theta_{t-1})$ (Get gradients w.r.t. stochastic objective at timestep t)
      $m_t \leftarrow (1 - 1/\tau_m) \cdot m_{t-1} + g_t/\tau_m$ (Update the *complementarily* modulated substances' density)
      $\rho_t \leftarrow \frac{\rho_{t-1} \cdot (\tau_\rho - 1) + \rho_{\text{rest}}}{\tau_\rho + \rho_{\text{rest}} \cdot |g_t|}$ (Update the *co-consumed* substances' density)
      $\theta_t \leftarrow \theta_{t-1} - \gamma m_t \rho_t$ (Update parameters/synaptic strengths)
   **end while**
   **return** $\theta_t$ (Resulting Parameters)

---

### 2.2.3 BIOLOGICALLY PLAUSIBLE REALIZATION OF RMSPROP

The second type of substances $\rho$ that is co-consumed by potentiation and depression stimuli is balanced by two factors: the intraneuron supplementation speed $s^+$, and the consumption speed $s^-$. First, we set the supplementation speed $s^+$ to be proportional to the difference between $\rho(t)$ and its resting value $\rho_{\text{rest}}$:

$$s^+(t) = k^+[\rho_{\text{rest}} - \rho(t)], \tag{10}$$

in which $k^+$ is a constant. Second, since $\rho$ is co-consumed during both potentiation and depression, we consider the rate of consumption for $\rho$ to be dependent on the absolute value $|g(t)|$ of the gradient. In addition, we make the consumption rate proportional to the current concentration. Consequently, the product of $|g(t)|$ and $\rho(t)$ sets the consumption rate:

$$s^-(t) = k^- \cdot \rho(t) \cdot |g(t)|, \tag{11}$$

where $k^-$ is also a constant. The overall dynamics of $\rho$ is controlled by $s^+$ and $s^-$:

$$\begin{aligned}
\tau_\rho \frac{d\rho(t)}{dt} &= s^+(t) - s^-(t) \\
&= k^+[\rho_{\text{rest}} - \rho(t)] - k^- \cdot \rho(t) \cdot |g(t)|.
\end{aligned} \tag{12}$$

**[Properties of Our Biological RMSProp]** Surprisingly, we show that at the equilibrium the dynamics in (12) converges to the desired quantity of $1/(\sqrt{v_t} + \epsilon)$ as what is used for weight update in the original RMSProp implementation (Tieleman & Hinton, 2012) based on (3). To see this, first note that at a constant gradient $g$ at equilibrium, the exponentially smoothed mean square root of gradient $\sqrt{v_t}$ converges to the absolute value of gradient $|g|$. So the RMSProp term converges to $1/(|g| + \epsilon)$. Furthermore, setting $d\rho(t)/dt = 0$ and $g(t) = g$ in (12) gives the equilibrium value of $\rho$ as:

$$0 = k^+[\rho_{\text{rest}} - \rho(\infty)] - k^- \cdot \rho(\infty) \cdot |g|$$
$$\Rightarrow \rho(\infty) = \frac{k^+ \rho_{\text{rest}}}{k^- |g| + k^+}, \tag{13}$$

which has a form very similar to that of the equilibrium value of the original RMSProp, $1/(|g| + \epsilon)$. We can force the equilibrium value $\rho(\infty)$ of the proposed dynamics to be $1/(|g| + \epsilon)$ by properly choosing constant values $k^+$, $k^-$, and $\rho_{\text{rest}}$ of our model:

$$\frac{k^+ \rho_{\text{rest}}}{k^- |g| + k^+} = \frac{1}{|g| + \epsilon} \Rightarrow \begin{cases} 1/\rho_{\text{rest}} = \epsilon \\ k^- = \rho_{\text{rest}} k^+ \end{cases} \tag{14}$$

Bring these two constrains into (12), we have:

$$\tau_\rho \frac{d\rho(t)}{dt} = [\rho_{\text{rest}} - \rho(t)] - \rho_{\text{rest}} \cdot \rho(t) \cdot |g(t)|, \quad \text{where } \rho_{\text{rest}} = 1/\epsilon \tag{15}$$

In the above, the constant $k^+$ is absorbed into the time constant $\tau_\rho$, and the hyperparameter $\tau_\rho$ sets the duration of the time window for the exponential moving average of $\rho$. Based on the same reasoning used to determine the value of $\tau_m$, we set $\tau_\rho = \frac{1}{1-\beta_2}$. For example, we set $\tau_\rho = 1000$ when $\beta_2 = 0.999$ is chosen in the Adam optimizer.

The proposed dynamics can be again discretized in time using the first-order forward Euler method:

$$\rho_t \leftarrow (1 - \frac{1}{\tau_\rho})\rho_{t-1} + \frac{1}{\tau_\rho} \cdot (\rho_{\text{rest}} - \rho_{\text{rest}} \cdot \rho_t \cdot |g_t|), \tag{16}$$

which simplifies to the following final rule for updating $\rho$:

$$\rho_t \leftarrow \frac{\rho_{t-1} \cdot (\tau_\rho - 1) + \rho_{\text{rest}}}{\tau_\rho + \rho_{\text{rest}} \cdot |g_t|}. \tag{17}$$

We conclude our proposed bio-plausible Adam optimizer (Bio-Adam) in Algorithm 1.

## 3 REALIZE WEIGHT SYMMETRY VIA PREDISPOSITION

### 3.1 PRELIMINARIES: THE WEIGHT TRANSPORT PROBLEM

It is generally believed that learning by backpropagation (BP) is not necessarily biological plausible in the brain for many reasons Stork (1989); Illing et al. (2019). One of the primary reasons is that it requires symmetrical weights between the forward and backward paths in order to propagate correct error information. This issue is also known as the weight transport problem Grossberg (1987); Ororbia II et al. (2017).

Using $\boldsymbol{x}^{(l+1)}$ and $\boldsymbol{y}^{(l+1)}$ to denote the input and output vectors of layer $(l+1)$ in an ANN, we have $\boldsymbol{y}^{(l+1)} = \sigma(\boldsymbol{x}^{(l+1)})$, where $\sigma$ is a non-linear activation function (Rumelhart et al., 1986). The input vector is given by:

$$\boldsymbol{x}^{(l+1)} = \mathbf{W}^{(l+1)}\boldsymbol{y}^{(l)} + \boldsymbol{b}^{(l+1)}, \tag{18}$$

where $\mathbf{W}^{(l+1)}$, $\boldsymbol{b}^{(l+1)}$ are the forward weights and bias in layer $(l+1)$, respectively. For a given loss function $L$, define error signals as $\boldsymbol{\delta}^{(l+1)} = \partial L/\partial \boldsymbol{x}^{(l+1)}$. BP propagates $\boldsymbol{\delta}^{(l+1)}$ to $\boldsymbol{y}^{(l)}$ by:

$$\boldsymbol{e}^{(l)} = \partial L/\partial \boldsymbol{y}^{(l)} = (\mathbf{W}^{(l+1)})^T \boldsymbol{\delta}^{(l+1)} \tag{19}$$

The backward weight $(\mathbf{W}^{(l+1)})^T$ is the transpose of the forward weights, which necessitates that the synaptic strength of the forward and backward routes be identical. This is considered biologically implausible for biological synapses that transmit signals in a single direction.

### 3.2 PROPOSED BIO-PLAUSIBLE WEIGHT SYMMETRY

#### 3.2.1 BIOLOGICAL MODELLING

Our bio-plausible setting is shown in Figure 2 (A). Following Lillicrap et al. (2016), we use a pair of synapses $w_{ij}$ and $w_{ji}$ between neuron $i$ and neuron $j$ to carry forward and backward signals, respectively. In addition, we employ the multi-compartments neuron model, which hypothesizes that neurons' basal and apical dendrites separately store two independent variables (Guerguiev et al., 2017). The basal dendrites receive the summed forward input signal $x$. In ideal situations, the apical dendrites receive the sum of top-down error signal $e = \partial L/\partial y$, which is the partial derivative of a given loss function $L$ with respect to the neuron's output $y$. Given the fact that the backward weight may not necessarily be equal to the forward weight, the top-down error signal may not be precisely equal to $\partial L/\partial y$, but rather provides only an estimation of it. The neuron processes the two stored variables $x$ and $e$ through its internal dynamics to generate the forward output $y = \sigma(x)$ to its next layer; and the backward error with respect to the input $\delta = \partial L/\partial x = e\sigma'(x)$ and propagates it to neurons in its previous layer. We refer interested readers to (Sacramento et al., 2018; Payeur et al., 2021) for more discussion of interneuron dynamics.

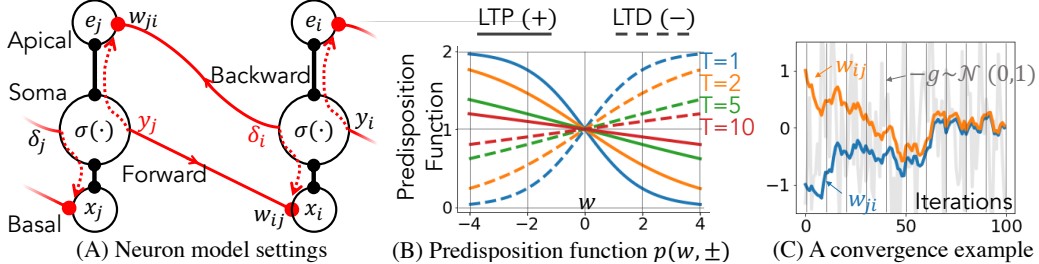

Figure 2: The model and the step function we use to realize weight symmetry. (A) The solid red lines represent axons and synapses that convey information in a single direction; the solid black lines represent dendrites that can transmit information bidirectionally through electrical spikes or molecular diffusion; and the dash red line represent our required variables sharing that helps build symmetrical weights. (B) An example series of stepsize function with different temperature parameter $T$. The solid line works for LTP while the dash line works for LTD. (C) Initialize two weights as $\pm 1$ respectively, and run gradient descent with stepsize = $0.1*p(w,\pm)$ and hyperparameter T=1.

### 3.2.2 PROPOSED BIO-PLAUSIBLE WEIGHT SYMMETRY LEARNING RULE

Under the above bio-plausible setting, there are two synaptic weights to be updated in learning. One is the weight $w_{ij}$ of the forward path synapse that connects to a basal dendrite of neuron $i$, and the other is the weight $w_{ji}$ of the backward path synapse that connects to an apical dendrite of neuron $j$. To facilitate successful learning, the adjusting goals of them are different: the forward weight $w_{ij}$ is updated by the top-down error signal directly to minimize a given loss function, while the backward weight $w_{ji}$ is updated to be symmetric with respect to its corresponding forward weight $w_{ij}$ to convey accurate error information.

Here, we discuss the proposed mechanisms that make the backward weight to be symmetric to the forward weight. The attainment of weight symmetry is accomplished in two stages. The first stage achieves the *establishment* of symmetry, i.e., aligning two initially different weights at the start of the learning process. The second stage deals with the *maintenance* of symmetry, which ensures that the values of forward and backward weights remain identical throughout the subsequent learning process. In the following, we first describe the maintenance of symmetry in the second stage assuming that the forward and reverse weights have already been aligned in the preceding first stage. Then, we elaborate on how we establish weight symmetry at the beginning of learning in the first stage.

[**Maintenance of Weight Symmetry by Gradient Transport**] Under the setting of biologically plausible multi-compartment based neural modeling, we may assume that the forward and backward synapses have access to the signals $y_j$ and $\delta_i$ locally. Note the product of these two signals is the gradient of the forward synaptic weight $w_{ij}$. Therefore, although the backward synapse with weight $w_{ji}$ cannot access $w_{ij}$, it may obtain the gradient of $w_{ij}$ based on the locally available signals $y_j$ and $\delta_i$. In other words, we employ *gradient transport* between the two synapses, which is considered biologically plausible (Akrout et al., 2019). Gradient transport allows the same gradient to be used in the weight updates of both the forward and backward synapses:

$$g_{ij}(t) = g_{ji}(t) = y_j(t)\delta_i(t). \tag{20}$$

With gradient transport, one can immediately see that for a BP method such as SGD, the amounts of weight change for the pair of forward and backward weights $w_{ij}$ and $w_{ji}$ are always identical. To see the same under the proposed Bio-Adam, we note that the momentum $m$ and the concentration $\rho$ of the co-consumed substances are integrated from the same gradient $g$, making them identical across the two synapses. Hence, the amounts of forward and backward weight changes mediated by Bio-Adam are also identical. As a result, both BP and Bio-Adam maintain the initially established weight symmetry between $w_{ij}$ and $w_{ji}$ since the amounts of update applied to the two weights are always identical.

[**Establishment of Weight Symmetry by Predisposition**] Building upon gradient transport, we achieve establishment of weight symmetry by further introducing a biologically plausible process

called predisposition. In this work, we specifically propose predisposition functions $p(w, \pm)$ that depend on two factors: synaptic strength $w$, and the direction of weight change (LTP or LTD).

Injecting $p(w, \pm)$ into the weight update rule of Bio-Adam in (5) results in a more comprehensive model of our bio-plausible synaptic dynamics:

$$w_t \leftarrow w_{t-1} - p(w_{t-1}, \text{sign}(-m_t)) \cdot \gamma \cdot m_t \cdot \rho_t. \tag{21}$$

Here, the negative momentum term $-m$ determines the updating type: $-m > 0$ corresponds to LTP, and $-m < 0$ correspond to LTD.

Now let us consider applying this rule to a pair of forward and backward weights $w_{ij}$ and $w_{ji}$. When an LTP update is applied to both $w_{ij}$ and $w_{ji}$, by predisposition we would like to potentiate the the weaker synapse of the two more than the other stronger one. As illustrated by the solid lines in Figure 2 (B), we achieve this predisposition effect by defining the predisposition function $p(w, +)$ in the form of the sigmoid function (Han & Moraga, 1995) with a hyperparameter $T$, which is usually considered as temperature :

$$p(w, +) = 2/(1 + exp(w/T)), \quad \text{(LTP)}, \tag{22}$$

Similarly, for a LTD update applied to $w_{ij}$ and $w_{ji}$, the desired predisposition effect would depress the larger weight more than the smaller one. As shown in the dashed lines in Figure 2 (B)), we achieve this by defining the predisposition function $p(w, -)$ according to:

$$p(w, -) = 2/(1 + exp(-w/T)), \quad \text{(LTD)}. \tag{23}$$

Under practical settings like small learning rate, we can see that the two predisposition effects designed above diminish the initial difference between $w_{ij}$ and $w_{ji}$ over time and eventually establish weight symmetry.

## 4 EXPERIMENTAL RESULTS

### 4.1 BIO-ADAM

Figure 3 compares the train/test loss curves of ANNs based on the widely adopted ResNet18 architectures (He et al., 2016) trained by four different optimizers on the well-known CIFAR10 dataset (Krizhevsky et al., 2009). Except for RMSProp, the performance of the remaining three optimizers on the test set is comparable. SGD+momentum performs the best on the train set, and our proposed Bio-Adam performs marginally better than Adam. RMSProp performs the worst.

In addition, we evaluated the wall-clock times of all four optimizers to determine the computing overhead Bio-Adam imposes. Running 200 training epochs on 4 NVIDIA RTX 3090 GPUs in parallel, Bio-Adam takes 55m52s, which is slightly slower than Adam's 54m07s; RMSProp's 53m15s; and SGD+Momentum's 52m56s.

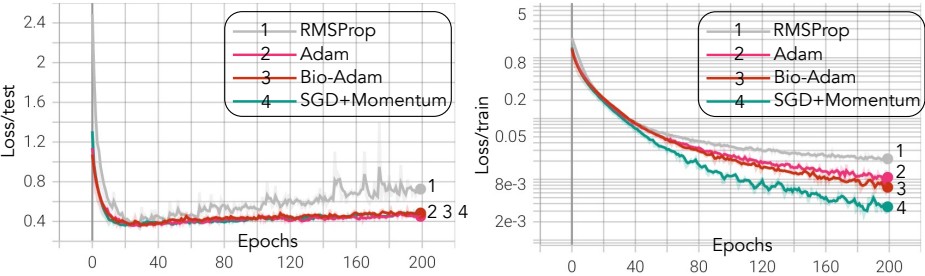

Figure 3: Cifar10 experiment. Network architecture: ResNet18 (He et al., 2016); All strengths of weight decay are set to zero; Learning rate $\gamma$=0.01 for RMSProp and SGD+momentum, $\gamma$=0.0001 for Adam and Bio-Adam; Detail setting for each optimizer: RMSProp: $\beta_2$=0.99, $\epsilon$=$10^-8$; Adam: $(\beta_1, \beta_2)$=(0.9, 0.999), $\epsilon$=$10^-8$; Bio-Adam: $(\tau_1, \tau_2)$=(10, 1000), $\rho_{rest}$=$10^8$; SGD+momentum: $\beta_1$=0.9.

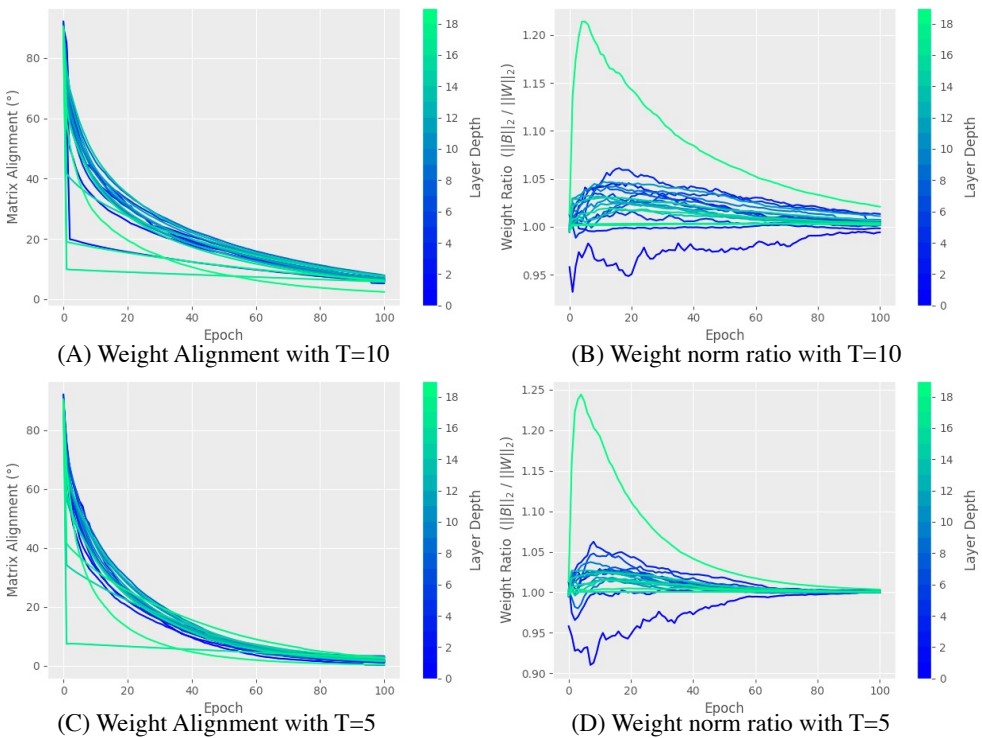

Figure 4: Cifar10 experiment. Network architecture: ResNet20 (He et al., 2016); Weight decay are set to zero; Learning rate $\gamma$=0.0001 for Bio-Adam with predisposition turned on. For (A) and (B), temperature T=10. For (C) and (D), temperature T=5

## 4.2 WEIGHT SYMMETRY

Throughout the training process, predisposition can establish and maintain the alignment between forward and backward paths. As demonstrated in Figure 4, we exhibit the alignment behavior of each layers individually in two forms: Sub-figures (A) and (C) evaluate the angle between the forward weight matrix $\mathbf{W}$ and the backward weight matrix $\mathbf{B}$ as:

$$\angle(\mathbf{B}, \mathbf{W}) = (180°/\pi)cos^{-1}(\mathbf{B} \cdot \mathbf{W}/(||\mathbf{B}||_2||\mathbf{W}||_2))$$

and Sub-figures (B) and (D) evaluate the weight norm ratio $(||\mathbf{B}||_2||/\mathbf{W}||_2)$.

Comparing different temperatures T, we can see that the lower temperatures establish symmetry more quickly. However, a temperature that is too low may hinder training performance. When comparing the (train/test) accuracy after the 100 epochs training, Adam reaches (99.17%/91.54%); Bio-Adam reaches (99.24%/91.53%); Bio-Adam + Predisposition (T=10) reaches (97.88%/91.55%), which is still comparable to Adam's. However, Bio-Adam + Predisposition (T=5) only reaches (95.31%/90.33%).

## 5 CONCLUSION

Based on synaptic dynamics, we present two ideas that, respectively, implement the bio-plausible Adam optimizer (Bio-Adam) and overcome the weight transport problem in BP. Bio-Adam eliminates the biologically problematic RMSProp term in favor of a synaptic substances dynamic $\rho$ that converges to the same equilibrium as RMSProp's. Meanwhile, Bio-Adam provides a clear biological explanation for the momentum term $m$. In addition, inspired by the predisposition property of synapses observed in neuroscience, we describe a novel method to build and maintain the symmetry between forward and backward paths using only local signals and without a separate training phase. The results of this investigation may impact future neuroscience research on the dynamics of synapses and on how they contribute to learning.

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

# A  ADDITIONAL EXPERIMENT

## A.1  TRAINING SPIKING NEURAL NETWORKS

### A.1.1  CIFAR10

Here, we train spiking neural networks (SNNs) (Yang et al., 2021) on the CIFAR10 dataset (Krizhevsky et al., 2009). SNNs are known for their noisy and sparse gradients due to spiking neuron's natural discontinuous 0/1 firing activity (Zenke & Ganguli, 2018; Shrestha & Orchard, 2018).

The experiment uses the recently proposed neighborhood aggregation (NA) direct training algorithm (Yang et al., 2021) to compute and propagate gradients. We use their open-source code (https://github.com/superrrpotato/Backpropagated-Neighborhood-Aggregation-for-Accurate-Training-of-Spiking-Neural-Networks) and only modify the optimizers to make a comparison between these four different optimizers: SGD+momenrtum, RMSProp, Adam and Bio-Adam.

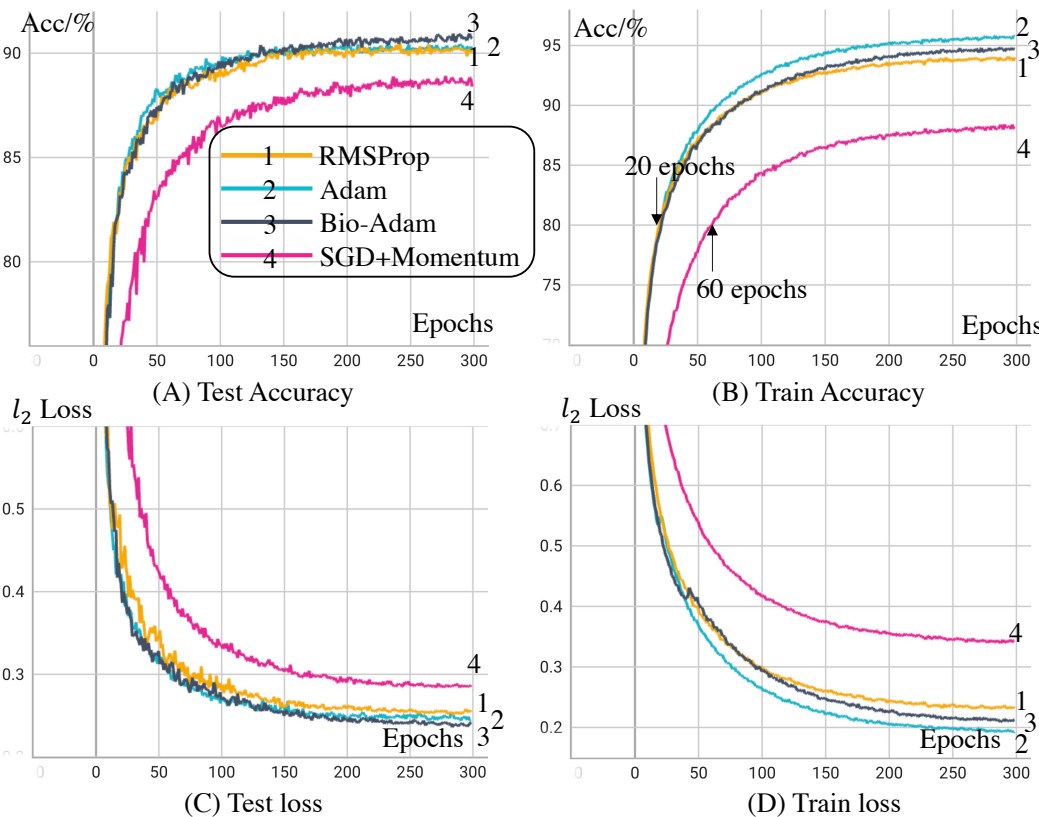

Figure 5: Cifar10 experiment on a spiking neural network trained by neighborhood aggregation (Yang et al., 2021). Network architecture: AlexNet 96C3-256C3-P2-384C3-P2-384C3-256C3-1024-1024 (iCj:convolution with depth i, kernel size j; Px:average pooling with kernel size x; 1024: fully connect layer with neurons number 1024); All strengths of weight decay are set to zero; Learning rate $\gamma$=0.05 for SGD+momentum, $\gamma$=0.0005 for Adam, Bio-Adam and RMSProp; Detail setting for each optimizer: RMSProp: $\beta_2$=0.99, $\epsilon$=$10^-8$; Adam: $(\beta_1, \beta_2)$=(0.9, 0.999), $\epsilon$=$10^{-8}$; Bio-Adam: $(\tau_1, \tau_2)$=(10, 1000), $\rho_{rest}$=$10^6$; SGD+momentum: $\beta_1$=0.9.

As shown in Figure 5, Bio-Adam, Adam and RMSProp reach comparable performance, where Bio-Adam has the marginally better testing accuracy and lower testing loss. Meanwhile, SGD+Momentum performs relatively poor, which confirms our conjecture about the advantages of different optimizers: Adam-liked optimizers has faster convergence when work with noisy/sparse gradient.

### A.1.2 ROBUSTNESS EVALUATION ON MNIST

In Figure 6, we empirically evaluate the robustness of the hyperparameters of our proposed Bio-Adam optimizer on the MNIST dataset (in solid lines). As a comparison, we also evaluate the performance of the vanilla Adam optimizer (in dashed lines).

As in the previous section, an SNN is trained using the NA algorithm (Yang et al., 2021). The x-axis represents the learning rate, and the y-axis represents the training loss in Figure 6 (A) (B) and represents the testing accuracy in Figure 6 (C) (D). All results are recorded after 10 epochs of training.

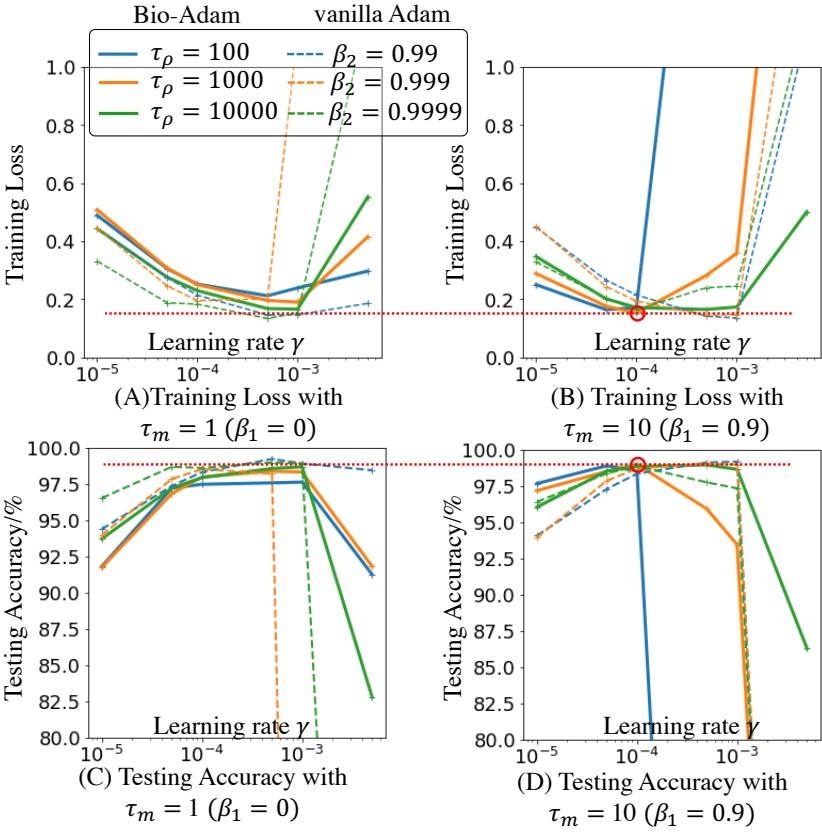

Figure 6: Network architecture:15C5-P2-40C5-P2-300 (iCj:convolution with depth i, kernel size j; Px:average pooling with kernel size x; 300: fully connect layer with neurons number 300); All strengths of weight decay are set to zero; $\rho_{rest} = 10^8$; The red circled data point is our default setting: $\tau_m = 10$, $\tau_\rho = 1000$, and $\gamma = 0.0001$. It is reported that the bias correction term in Adam boosts its robustness at the start of training (Kingma & Ba, 2014). However, biology learning is a lifelong process. Moreover, the bias correction terms appear to lack a clear biological correspondence. Due to these two considerations, we omitted them from this experiment in order to make a fair comparison.

In Figure 6, Different combinations of $\tau_m$ and $\tau_\rho$ can achieve equivalent performance with a suitable choice of the learning rate $\gamma$, demonstrating the robustness of the hyperparameters. In the meantime, the default configuration (red circled) derived from Adam's default setting achieves the lowest training loss and the highest testing accuracy among all Bio-Adam's results ($(\beta_1, \beta_2) = (0.9, 0.999)$ in Adam corresponds to $\tau_m = 10$, and $\tau_\rho = 1000$ in Bio-Adam; and adopt the same default learning rate $\gamma = 0.0001$). When compared to vanilla Adam, Bio-Adam gains comparable performance in both accuracy and robustness.

