# OpenReview forum: "Synaptic Dynamics Realize First-order Adaptive Learning and Weight Symmetry"
_ICLR.cc/2023/Conference — Submitted to ICLR 2023_

### Official Review · Reviewer_hKeF · 2022-10-25

**Confidence:** 4
**Correctness:** 2
**Technical Novelty And Significance:** 2
**Empirical Novelty And Significance:** 2
**Recommendation:** 5

**Clarity, Quality, Novelty And Reproducibility:**

The paper is well written and easy to follow. However, its current originality is limited based on my current view.

**Strength And Weaknesses:**

Strength:
The idea of explaining the gradient update from neuroscience view is interesting.

Weakness:
1. In terms of equation (6) and (7), it seems to be a very deliberate operation to split the concentration and gradient into both positive and negative parts here. What do you refer to by the $m^+$ and $m^-$ here? The high flux or low concentrations of $Ca^{2+}$? Why can you assume that it is depending on the "gradient" with a linear relation? For me, there lacks some evidence here to support the reasoning procedure beyond assumption. For example, ref [1] gives the dynamics equation for LTP/LTD.

2. The model for RMSProp has the same issue as above. Rewrite the existing updating rule into a flow is not convincing as you can associate the parameter values with high arbitrariness. I expect that there should be some intermediate process that explains why it takes the specific parameters.


Ref:
[1] https://www.pnas.org/doi/epdf/10.1073/pnas.132651299

**Summary Of The Paper:**

In this paper, the authors propose an interpretation of the first-order gradient model based on the dynamic evolution of synapse. Overall, it is interesting but the explanation itself seems to be not based on the ground truth of neuroscience thus appears to be a bit post-hoc given the establishment of ADAM and RMSProp.

**Summary Of The Review:**

Overall, I think this is an interesting paper with an promising view to explain the gradient descent model with biological mechanism. However, there lacks some rigorous and detailed reasoning on how they are actually connected to each other.

---

> ### Author Response · Authors · 2022-11-08
> **Response to reviewer hKeF (part 1)**
>
> Thanks very much for your review and your recognition of our idea to interpret the first-order gradient models based on the dynamic evolution of synapse. We want to explain and clarify to your comments piece by piece as followed:
> ***
> Q1: In terms of equation (6) and (7), it seems to be a very deliberate operation to split the concentration and gradient into both positive and negative parts here. What do you refer to by the m+ and m- here? The high flux or low concentrations of Ca2+?
> A1: Sorry for this confusion. We decompose Adam’s momentum term m into two separate dynamics, each of which has clear chemical correspondence and can be positive valued. As explained in the 5th paragraph of Sec 2.2.1, m+ and m- refer to the density of substances that modulated by complementary processes like phosphorylation and de-phosphorylation apply such as calcium/calmodulin-dependent kinase II (CaMKII), cAMP response element-binding protein (CREB) and cofilin. The substances in phosphorylated form are named as m+, and the substances in de-phosphorylated form are named as m-. This follows how LTP/LTD defined in neuroscience works like [1], and our m+ and m- are abstract variables to model the similar behaviors that take a role in the learning process, where m+ induces LTP and m- induces LTD.
> ***
> Q2: Why can you assume that it is depending on the "gradient" with a linear relation?
> A2: We follow previous works settings where linear leaky integration is adopted to model the dynamics of substances density [2] and to model many other biological processes like membrane potential and postsynaptic current [3]. Meanwhile, our modeling target, the reversible phosphorylation, which is m+ and m- in our work, plays a critical role in binding of effector proteins. As modeled in [4], a linear dynamics model is used to describe such binding dynamics. The gradient signal is assumed to be readily available in this work, and our contribution is to build upon it and to describe the possible biological synapses’ dynamics in realizing the Adam-liked learning. [5][6] discuss the generation and propagation of gradient, and we acknowledge that this is an ongoing research direction.
> ***
> Q3: For me, there lacks some evidence here to support the reasoning procedure beyond assumption. For example, ref [1] https://www.pnas.org/doi/epdf/10.1073/pnas.132651299 gives the dynamics equation for LTP/LTD.
> A3: Thanks very much for your constructive advice, and we agree these important biological reasoning should be comprehensively introduced in the paper as I described in A1 and A2. This suggestion will not only benefit for this revised version but also our future papers. Additionally, we go through the paper you mentioned carefully: The paper proposed a two-component model. Each component follows a first order leaky integration dynamics with different time constant, which is a very similar modeling choice as in our setting. The two components describe certain integrated properties of pre- and post-synaptic potentials. With a proper choice of parameters, the work successfully restores the well-known STDP and BCM theories. We admire this work in connecting their modeling to the well-observed existing synapse plasticity rules when synapses are stimulated by a constant frequency. However, our motivation comes from the issue of how synapses may synthesis the learning signals that are constantly changing in a relatively long period of time. Instead of explaining the existing biology phenomenon, we provide a hypothesis for experimental neuroscience to explore. We demonstrate our biological evidence in the molecular level. In this perspective, we think our evidence would be sufficient.
> ***
> [1] Dendritic spine loss and synaptic alterations in Alzheimer’s disease. Molecular neurobiology 08
> [2] Neuromodulator-dependent synaptic tagging and capture retroactively controls neural coding in spiking neural networks. Scientific reports 22
> [3] Spiking neuron models: Single neurons, populations, plasticity. Cambridge university press 02
> [4] Dynamic impact of temporal context of Ca2+ signals on inhibitory synaptic plasticity. Scientific reports 11
> [5] Burst-dependent synaptic plasticity can coordinate learning in hierarchical circuits. Nature neuroscience 21
> [6] Dendritic cortical microcircuits approximate the backpropagation algorithm. NeurIPS 18

---

> > ### Comment · Reviewer_hKeF · 2022-11-13
> > **Concerns w.r.t to the Biological Motivation**
> >
> > Thanks for the quick response to my questions and additional explanations on the confusing part.
> >
> > I understand the authors' response to the motivation. However, my major concerns here remain here from two aspects. First, the model is still like a posterior interpretation rather than a mechanism-driven brain-inspired algorithm. Second, although the authors provided references of why they set up the paradigm in its current form, it is still not a widely accepted or appreciated model in neuroscience. Thus I will keep my views on these points.

---

> > > ### Author Response · Authors · 2022-11-13
> > > **2nd round response to reviewer hKeF**
> > >
> > > Many thanks for your response. Our bioplausible vision is based on molecular-level mechanisms, such as the balance of phosphorylation and de-phosphorylation, which corresponds to the momentum term $m$; and the slow chemical/biological processes, such as translation and transcription, which correspond to the RMSProp term $\rho$. Calcium/calmodulin-dependent kinase II (CaMKII), cAMP response element-binding protein (CREB), and cofilin explain $m$; Brain Derived Neurotrophic Factor (BDNF) explains $\rho$. These are all key molecules in plastic changes. Unfortunately, it fails to convince you.
> > >
> > > To validate our objective, we wish to present you with more prior research in bio-plausible BP, which can also be considered as bridging the gap between AI and neuroscience from a posterior interpretation. BP necessitates that each neuron simultaneously store activation and error signals, therefore [1] presented a multi-compartment neuron model to store the two signals independently. In order to achieve bio-plausible learning without the forward/reverse phases of BP, [2] introduced a microcircuit that couples each Pyramidal cell with an SST cell based on the predictive coding algorithm. FA and DFA are presented in [3] and [4] to circumvent BP's bio-implausible forward-backward synapses correspondence. All of these works are the result of BP's success. From there, people consider whether the brain is capable of implementing BP and, if not, how we may adapt BP to make it more biologically plausible. Similarly, our work is motivated by the success of modern optimizers such as Adam, which leads us to consider which aspects of it are bio-implausible and whether or not there are biological mechanisms that could conceivably support it.
> > >
> > > Another point of view is that studying the learning mechanism in the real brain is never easy. When compared to the recent progress of AI, the existing fully neuroscience-observed biology mechanisms such as STDP or BCM remain quite naive and frequently fail in more complex learning tasks [5][6]. As a result, rather than waiting for complete comprehension in neuroscience, the development of brain-inspired algorithms may yield some hypothetical notions to be evaluated by neuroscience instead.
> > > ***
> > > [1] Towards deep learning with segregated dendrites. Elife, 2017.
> > > [2] Dendritic cortical microcircuits approximate the backpropagation algorithm. NeurIPS, 2018.
> > > [3] Random synaptic feedback weights support error backpropagation for deep learning. Nature communications,2016.
> > > [4] Direct feedback alignment provides learning in deep neural networks, NeurIPS 2016.
> > > [5] Supervised learning in spiking neural networks with ReSuMe: sequence learning, classification, and spike shifting[J]. Neural computation, 2010.
> > > [6] A biologically plausible supervised learning method for spiking neural networks using the symmetric STDP rule. Neural Networks, 2020.

---

> ### Author Response · Authors · 2022-11-08
> **Response to reviewer hKeF (part 2)**
>
> Q4: The model for RMSProp has the same issue as above.
> A4: Thanks for your constructive advice. We put the focus of this paper in describing the algorithm instead of biological process and we are very sorry for this neglect. For RMSProp, our designed dynamics of $\rho$ refers to non-instantaneous processes such as slow chemical reactions and biological processes like translation and transcription. For example, the Brain Derived Neurotrophic Factor (BDNF), a key molecule involved in plastic changes. ProBDNF preferentially binds p75 NTR receptor, which facilitates LTD. On the other hand, BDNF in its mature form binds specifically to tyrosine kinase receptors and facilitates LTP [7]. In other words, both LTD and LTP consumes BDNF. The supplementation of BDNF requires the transcription of new mRNAs in the nucleus [8]. Transcription is slow as compares to the momentum term’s phosphorylation process (Interestingly, when we choose the parameters correspond to empirically tuned Adam’s parameter, we also have $\tau_m=10$ and $\tau_\rho=1000$). We will discuss the biological reasoning of both Momentum and RMSProp in detail in the revised version. Thanks for your constructive suggestion again.
> ***
> Q5: Rewrite the existing updating rule into a flow is not convincing as you can associate the parameter values with high arbitrariness. I expect that there should be some intermediate process that explains why it takes the specific parameters.
> A5: Sorry for this misunderstanding. We are not simply “rewrite into a flow”. Our model has two major differences from the original: 1. Adam is an empirical learning rule, while our rule is completely based on Biological plausible dynamic processes. 2. Structurally, It’s not trivial to reform Adam into a bio-plausible flow. The first contribution is to find the bio-plausible mechanisms and corresponding molecular evidences. The second is to derive the same equilibrium, which guarantees the comparable performance to Adam. Particularly, we consider the derivation of bio-plausible rules of the RMSProp term in eq(17) as a solid contribution. The rule is based on our defined bio-plausible meaning without using square root or division. Meanwhile, it surprisingly can reach the same forms of equilibrium with the original one.
> ***
> Q6: as you can associate the parameter values with high arbitrariness. I expect that there should be some intermediate process that explains why it takes the specific parameters.
> A6: Our algorithm provides a more general bio-plausible framework with parameters free to choose. The framework can be reduced to the Adam, which is commonly used in the practice for training ANN, if our parameters are chosen according to the rules described in eq(14). In this way of choosing parameters, we can make a fair comparison experimentally in the image classification tasks. It is possible for experimental neuroscience to measure the time constant of each substance and answer what is our brain’s choice.
> ***
> [7] Brain-Derived Neurotrophic Factor: A Key Molecule for Memory in the Healthy and the Pathological Brain. Front. Cell. Neuroscience 19
> [8] Extracellular Metalloproteinases in the Plasticity of Excitatory and Inhibitory Synapses. Cells 21

---

> > ### Comment · Reviewer_hKeF · 2022-11-13
> > **Concerns on computational efficacy**
> >
> > Thanks for answering my questions and providing additional results to support the claim of model superiority.  To make the whole story coherent, it is necessary here to provide evidence here that the proposed optimization pipeline can achieve "better" performance compared to the generally adopted ones either on ANN or SNN training. For this purpose, CIFAR-10 for ANN is not sufficient. For ANNs, I think the training on ImageNet at least with EfficientNet is necessary to convince the readers if no transformer-family methods are included. For SNN, additional experiments on CIFAR-100, DVS-CIFAR, or ImageNet should also be included with the recent progression of SNN training pipelines (The references are easy to find here and I do not plan to list them in order to keep the anonymous requirement).
> >
> > As the theoretical proof may be difficult, I won't strongly require that. If the authors can provide more intuitive/visual explanation on why it works with sufficient evidence, I am willing to increase my scores here.

---

### Official Review · Reviewer_2dmD · 2022-10-26

**Confidence:** 3
**Clarity, Quality, Novelty And Reproducibility:** The paper is clear and of high quality.
**Correctness:** 3
**Technical Novelty And Significance:** 3
**Empirical Novelty And Significance:** 2
**Recommendation:** 5

**Strength And Weaknesses:**

Strengths
- Understanding how the brain achieves credit assignment is an important and very interesting research direction. This paper contributes to this field and provides a hypothesis for experimental neuroscience to explore.
- Bio-adam is an interesting and novel algorithm.
- The paper is well written and clear.

Weaknesses
- The main contribution of this work is to present a biologically motivated implementation of adam (there are no computational advantages proposed). However, in my view the biological connections are undeveloped, and only loosely motivated by neurobiology. As a result I find this work’s contributions limited.
- I do not find gradient transport and the weight symmetry learning rule particularly biologically plausible, and as such do not find this aspect of the work that convincing. First, there is little to no evidence for 1 to 1 forward backward connections in the brain being a general principle. Second, backward synapses would need to be able to communicate signed errors, which they cannot, or alternatively there should be two sets of backward synapses for each sign. Third, in my understanding axonal signaling is not equipped to signal with the time resolution that would be required.
- There is no advantage of bio-adam shown over a more simple update such as sgd+momentum, which would most likely be simpler for biological systems to approximate.


**Summary Of The Paper:**

This work presents a biologically motivated implementation of ADAM and weight transport solution.


**Summary Of The Review:**

Although interesting and novel, there are no computational advantages presented and the biological connections are approximate. Therefore I find this work’s contributions limited.

---

> ### Author Response · Authors · 2022-11-08
> **Response to reviewer 2dmD**
>
> Thanks for your time in reviewing our paper. Your assessments and questions give us many insights. Here, we want to provide you some different perspectives to discuss further.
> ***
> Q1: I do not find gradient transport and the weight symmetry learning rule particularly biologically plausible, and as such do not find this aspect of the work that convincing.
> A1: Biological brains incorporate many learning principles that are not well understood. Like many related works, we attempt to address the bio-plausibility of some of those computational principles people hypothesized. There has been existing works [1][2] suggesting the bio-plausible of gradient transport. Meanwhile, many prior works recognize weight symmetry as biological implausible. Within the scope of this work, our key focus is to develop new bio-plausible Adam like learning principles; Based on the existing arguments of the bio-plausibility of gradient transport, we address the bio-plausibility of the weight symmetry problem.
> ***
> Q2: There is little to no evidence for 1 to 1 forward backward connection in the brain being a general principle.
> A2: Yes, we agree that it is not well understood how forward and backward connections are formed in the brain in relation to learning. Here, we want to provide you more prior works in addressing the bio-plausibility of the credit assignment problem. The majority of existing works assume a one-to-one mapping [2] [3] [4]. With the assumption of 1 to 1 forward backward connection, [2] proposed the possibility to convey error information through spike burst. [3] presented that the 1 to 1 connection with random backward weights can facilitate learning. [4] proposed a hypothetical predictive coding microcircuit that a SOM cell cancels its pairing pyramidal cell’s 1 to 1 backward bias current and leaves clean error information on the previous layers’ neurons’ apical dendrites. In general, we concur that this may be a simplification. This is also one of the underlying assumptions of our work.
> ***
> Q3: Backward synapses would need to be able to communicate signed errors, which they cannot, or alternatively there should be two sets of backward synapses for each sign.
> A3: Thanks for your constructive advice. We agree with you that our model could be made more biological plausible on this problem. We could utilize two sets of synapses of different signs with respecting to Dale's law, or transmit information of both signs by positive only spikes by setting the threshold between negative and positive effects as a positive number (Low strength induces LTD and high strength induces LTP as observed in the BCM theory [5]), or use spike burst multiplexing like [2] to convey forward and backward signals together. This would **not change** the core ideas of implementing this Adam like dynamics, but would kind of increase the complexity, which will be explored in our future works. Your suggestion will improve the bio-plausibility of our work.
> ***
> Q4: In my understanding axonal signaling is not equipped to signal with the time resolution that would be required.
> A4: We adopt similar continuous time and real valued setting as previous works [2][4]. Both [2] and [4] follows the setting that data is feeded-into the network continuously, and synapses use the real time generated error signals to update their weights. The real valued forward backward signals stand for spike firing-rates of biological neurons.
> ***
> Q5: There is no advantage of bio-adam shown over a more simple update such as sgd+momentum, which would most likely be simpler for biological systems to approximate.
> A5: In the CIFAR10 experiment we prove the comparable performance between Bio-Adam and Adam. Indeed, sgd+momentum performs the best in this experiment. However, Adam liked optimizers have advantages over sgd+momentum on many other tasks. To prove this, we add a new experiment to train on spiking neural networks (known for noisy and sparse gradient due to spiking neuron's natural discontinuous 0/1 firing activity) on CIFAR10 [6] and compare the performance of these four different optimizers. In the result, Bio-Adam, Adam and RMSProp have comparable performance, and clearly outperform sgd+momentum. For example, they reach 80% training accuracy 3x faster than sgd+momentum. Please see the appendix section of the revised file for detail.
> ***
> [1] Deep learning without weight transport. NeurIPS 19
> [2] Burst-dependent synaptic plasticity can coordinate learning in hierarchical circuits. Nature neuroscience 21
> [3] Random synaptic feedback weights support error backpropagation for deep learning. Nature communications 16
> [4] Dendritic cortical microcircuits approximate the backpropagation algorithm. NeurIPS 18
> [5] Theory for the development of neuron selectivity: orientation specificity and binocular interaction in visual cortex. Journal of Neuroscience 1982
> [6] Backpropagated Neighborhood Aggregation for Accurate Training of Spiking Neural Networks. ICML 21

---

### Official Review · Reviewer_eRea · 2022-11-07

**Confidence:** 2
**Clarity, Quality, Novelty And Reproducibility:** The paper is clearly written and good…
**Correctness:** 3
**Technical Novelty And Significance:** 3
**Empirical Novelty And Significance:** 2
**Recommendation:** 6

**Strength And Weaknesses:**

Strengths
---------------
The idea of biological neural systems implementing an adaptive gradient optimization procedure is an interesting and novel contribution to our knowledge of learning in neural circuits.

Weaknesses
----------------

1. There are spelling errors:
    (A) “Variable” on first line on Page 2
    (B) “Mechanism” on  second line of last paragraph in Introduction (Page 2)
2. The RMSProp bio-plausible realization is somewhat arbitrary and underdeveloped. The paper doesn’t provide the corresponding biological substance that is co-consumed by potentiation and depression. This justification was provided for momentum.
3. The empirical experiments are inconclusive. Evaluating Bio-Adam on a single architecture (ResNet18) is not a sufficient justification for how the method compares to its contemporaries. This makes me question the robustness of the hyperparameters and the Bio-Adam model.



**Summary Of The Paper:**

The authors propose a bio-plausible realization of the Adam algorithm in biological neural systems. They similarly propose a method to establish weight symmetry in biological neural networks based on the concept of predisposition.

**Summary Of The Review:**

The bio-plausible realization of ADAM is an interesting contribution to the learning dynamics of neural circuits. My main criticism of this work is the empirical experiments conducted in this paper. The experiments are not comprehensive, more work can be done to empirically evaluate Bio-Adam.

I am willing to revise my score if more empirical experiments are conducted.

---

> ### Author Response · Authors · 2022-11-09
> **Response to reviewer eRea**
>
> Thanks for your time in reviewing our paper and we are gratitude to have your recognition of our work. We will fix all writing glitches, improve clarity and quality of writing, correct the confusions with the major issues responded below:
> ***
> Q1: The RMSProp bio-plausible realization is somewhat arbitrary and underdeveloped. The paper doesn’t provide the corresponding biological substance that is co-consumed by potentiation and depression. This justification was provided for momentum.
> A1: Thanks very much for your kind reminder. we are very sorry for this neglect. Reviewer hKeF presented the same concern and we has provided the response in A4 for him/her. But we are very sorry for this neglect for you again. We put the focus of this paper in describing the algorithm instead of biological process and we are very sorry for this neglect. For RMSProp, our designed dynamics of $\rho$ refers to non-instantaneous processes such as slow chemical reactions and biological processes like translation and transcription. For example, the Brain Derived Neurotrophic Factor (BDNF), a key molecule involved in plastic changes. ProBDNF preferentially binds p75 NTR receptor, which facilitates LTD. On the other hand, BDNF in its mature form binds specifically to tyrosine kinase receptors and facilitates LTP [1]. In other words, both LTD and LTP consumes BDNF. The supplementation of BDNF requires the transcription of new mRNAs in the nucleus [2]. Transcription is slow as compares to the momentum term’s phosphorylation process (Interestingly, when we choose the parameters correspond to empirically tuned Adam’s parameter, we also have $\tau_m=10$ and $\tau_\rho=1000$). We have included this part in the revised version. Thanks for your constructive suggestion again.
> ***
> Q2: The empirical experiments are inconclusive. Evaluating Bio-Adam on a single architecture (ResNet18) is not a sufficient justification for how the method compares to its contemporaries. This makes me question the robustness of the hyperparameters and the Bio-Adam model.
> A2: I am sorry for the lack of experiments. Reviewer 2dmD also questioned why we need bio-plausible Adam if the simpler SGD+momentum performs the best in the ResNet18, CIFAR10 experiment. We have included two additional experiments in the appendix section of our revised version. Firstly, we train an AlexNet structured spiking neural network (SNN) on the CIFAR10 dataset. We choose to compare on SNNs because SNNs are known for their noisy and sparse gradient due to the spiking neuron’s all or none firing activity. Therefore, training on SNNs can better show the advantage of Adam-liked optimizers. In the experiment results, Bio-Adam, Adam and RMSProp gain comparable performance, whereas SGD+momentum performs relatively poor. Secondly, to further test Bio-Adam’s robustness, we empirically evaluate the effect of different hyperparameters on the MNIST dataset. In the result, different combinations of $\tau_m$ and $\tau _{\rho}$ can achieve equivalent performance with a suitable choice of the learning rate $\gamma$, demonstrating the robustness of the hyperparameters. In the meantime, the default configuration derived from Adam's default setting achieves the lowest loss value ($(\beta_1,\beta_2)=(0.9,0.999)$ in Adam corresponds to $\tau_m=10$, and $\tau_\rho=1000$ in Bio-Adam; and adopt the same default learning rate $\gamma =0.0001$). Please check the appendix section of the revised paper for more detail.
> ***
> We really appreciate it if you can kindly re-check our insights and additional experiments and re-consider your ratings.
> ***
>
> [1] Brain-Derived Neurotrophic Factor: A Key Molecule for Memory in the Healthy and the Pathological Brain. Front. Cell. Neuroscience 19
> [2] Extracellular Metalloproteinases in the Plasticity of Excitatory and Inhibitory Synapses. Cells 21

---

> > ### Comment · Reviewer_eRea · 2022-11-16
> > **Empirical experiments**
> >
> > Thank you for making the effort to address my concerns. I have acknowledged the changes in the paper, specifically the paragraph justifying the co-consumption of LTD and LTP, and similarly the empirical experiments in the appendix section. I can also appreciate the choice behind evaluating Bio-Adam on spiking neural networks due to its bio-plausible spiking activity.
> >
> > With regards to the robustness evaluation on MNIST, it would be more beneficial to observe the test accuracy as a function of the learning rate. Loss function values can occasionally be misleading, as there are occasions where the loss increases due to overfitting. Could you plot the accuracy and compare this with vanilla Adam (side by side). I realize that this might be too much considering that the deadline is soon approaching but I think this would improve the manuscript.

---

> > > ### Author Response · Authors · 2022-11-17
> > > **2nd round response to reviewer eRea**
> > >
> > > It is never too late to get excellent advice, so don't worry about the deadline. We are very grateful to meet such a responsible reviewer. We have taken your suggestion seriously and added the test accuracy in the robustness experiment. In addition, vanilla Adam is also included to make a comparison.
> > > Considering your recognition of our work, we really appreciate it if you can kindly re-check our revised manuscript. We'd like to know if you have any other questions. We would be very grateful that if you can re-consider your rating on the premise that we could address your concerns with our best efforts.

---

### Author Response · Authors · 2022-11-11
**Hoping to discuss with the reviewers**

Dear Chairs and Reviewers,

We are writing this letter to show our hope of having discussions with you, according to your comments and concerns of our work. There are some misunderstandings in these concerns and comments, which would lead to a misjudgment of our work. We treated these concerns carefully and had made efforts to explain them. Could you please take a little bit of time to check up our responses and discuss with us? Thank you all!

Best regards.

---

### Decision · Program_Chairs · 2023-01-20

**Decision:**

Reject

**Justification For Why Not Higher Score:**

This paper unfortunately does not provide sufficient experimental evidence to prove that the proposed bio-Adam algorithm works well. The proposed algorithm seems a bit posthoc as well. Only one reviewer was leaning towards acceptance and they had a weak confidence score.

**Justification For Why Not Lower Score:**

N/A

**Metareview: Summary, Strengths And Weaknesses:**

### Summary
Gradient based adaptive learning rule algorithms such as Adam and RMSProp have been instrumental at achieving state of art deep learning results. This paper aims to develop a biologically-plausible variant of Adam. The proposed learning rule can run in continuous time and works with spiking neural networks. In this paper, they show comparable results to Adam with a more biologically plausible algorithm.

Below I will summarize some of the strengths and the weaknesses of the paper as pointed out by the reviewers:

### Strengths

- Understanding how the brain achieves credit assignment is an important research direction.
- The proposed biologically plausible Adam is a novel algorithm.
- The paper is well-written and easy-to-follow.

### Weaknesses

- The experiments are not comprehensive enough. More work needs to be done to empirically evaluate Bio-Adam.
- The biological connections are not developed well enough. The method is only loosely motivated by neurobiology.
- The proposed model i like a posterior interpretation rather than a mechanism-driven brain-inspired algorithm since we know that Adam works well for DL. It feels like an adaptation to Neuroscience.
- The model bio-Adam is based from is not a widely accepted paradigm in neuroscience.

### Decision

The subject this paper is studying is an interesting one and the proposed algorithm is novel. However, only one of the reviewer of this paper was leaning towards acceptance and with a weak confidence level. The weaknesses pointed out by the reviewers in terms of experimentations and motivations indicate that this paper is not ready for publication yet.